# BCC0 collaborates with IMC32 and IMC43 to form the *Toxoplasma gondii* essential daughter bud assembly complex

**Rebecca R. Pasquarelli**[1], **Jihui Sha**[2], **James A. Wohlschlegel**[2], **Peter J. Bradley**[1,3]*

**1** Molecular Biology Institute, University of California, Los Angeles, California, United States of America, **2** Department of Biological Chemistry and Institute of Genomics and Proteomics, University of California, Los Angeles, California, United States of America, **3** Department of Microbiology, Immunology, and Molecular Genetics, University of California, Los Angeles, California, United States of America

* pbradley@ucla.edu

**Data Availability Statement:** All relevant data are within the manuscript and its Supporting Information files.

## Abstract

*Toxoplasma gondii* divides by endodyogeny, in which two daughter buds are formed within the cytoplasm of the maternal cell using the inner membrane complex (IMC) as a scaffold. During endodyogeny, components of the IMC are synthesized and added sequentially to the nascent daughter buds in a tightly regulated manner. We previously showed that the early recruiting proteins IMC32 and IMC43 form an essential daughter bud assembly complex which lays the foundation of the daughter cell scaffold in *T. gondii*. In this study, we identify the essential, early recruiting IMC protein BCC0 as a third member of this complex by using IMC32 as bait in both proximity labeling and yeast two-hybrid screens. We demonstrate that BCC0's localization to daughter buds depends on the presence of both IMC32 and IMC43. Deletion analyses and functional complementation studies reveal that residues 701–877 of BCC0 are essential for both its localization and function and that residues 1–899 are sufficient for function despite minor mislocalization. Pairwise yeast two-hybrid assays additionally demonstrate that BCC0's essential domain binds to the coiled-coil region of IMC32 and that BCC0 and IMC43 do not directly interact. This data supports a model for complex assembly in which an IMC32-BCC0 subcomplex initially recruits to nascent buds via palmitoylation of IMC32 and is locked into the scaffold once bud elongation begins by IMC32 binding to IMC43. Together, this study dissects the organization and function of a complex of three early recruiting daughter proteins which are essential for the proper assembly of the IMC during endodyogeny.

## Author summary

*Toxoplasma gondii* is an obligate intracellular parasite that causes severe and even fatal disease in congenitally infected neonates and the immunocompromised. The parasite replicates using an internal budding mechanism called endodyogeny, in which two daughter buds form within the cytoplasm of a maternal parasite. Nascent daughter buds are scaffolded by the inner membrane complex (IMC), a unique organelle found in *T. gondii* and

**Funding:** This work was supported by NIH grants AI123360 to P.J.B. and GM153408 to J.A.W. R.R.P was supported by the Ruth L. Kirschstein National Research Service Awards GM007185 and AI007323, as well as the UCLA Molecular Biology Institute Whitcome Fellowship. The funders had no role in study design, data collection and analysis, decision to publish, or preparation of the manuscript.

**Competing interests:** The authors have declared that no competing interests exist.

other parasites such as *Plasmodium spp.*, the causative agent of malaria. Many proteins that localize to the IMC of daughter buds have been identified, but only three of these proteins are essential for parasite replication and survival: IMC43, IMC32, and BCC0. In this study, we demonstrate that these three proteins exist in a complex, explore which features of BCC0 are critical for its localization and function, and develop a model for how the complex is assembled. This work expands our understanding of how the foundation of the early daughter buds is established during endodyogeny.

## Introduction

*Toxoplasma gondii* is a protozoan pathogen that can infect any warm-blooded animal worldwide [1]. *T. gondii* belongs to the Apicomplexa, a phylum of obligate intracellular parasites that cause significant human morbidity and mortality including *Plasmodium spp.* (malaria) and *Cryptosporidium spp.* (diarrheal disease) [2,3]. The phylum also includes important veterinary pathogens such as *Eimeria spp.* (chicken coccidiosis) and *Neospora caninum* (neosporosis) which cause economic loss in the livestock industry [4–6]. Approximately 30% of the global human population is chronically infected with *T. gondii* [7]. Healthy individuals typically remain asymptomatic, but infection can result in severe or even fatal disease in immunocompromised individuals or congenitally-infected neonates [8–10]. Current treatments can limit the acute disease but cannot clear the parasite from the host, resulting in lifelong chronic infection [9]. To identify targets for the development of new therapeutics, a deeper understanding of the parasite's unique biology is needed.

One unique feature of apicomplexan parasites is the inner membrane complex (IMC), a specialized organelle that plays critical roles throughout the *T. gondii* lytic cycle [11]. The *T. gondii* IMC underlies the parasite's plasma membrane and has both membrane and cytoskeletal components. The first layer is a series of flattened vesicles called alveoli which are sutured together in a quilt-like pattern [12]. Many IMC-localizing proteins associate with the membranes of the alveoli using either transmembrane domains or acylation [13–17]. The second layer of the IMC is composed of intermediate filament-like proteins called alveolins which form a supportive mesh-like network [18,19]. Additional IMC proteins associate with the network by binding to the alveolins [20,21]. Within the IMC, there are three distinct subdomains that have different protein components and functions. At the apex of the parasite, a single cone-shaped vesicle defines the apical cap. Proteins that localize specifically to this region of the organelle such as AC9 and AC10 are required for host cell invasion [21,22]. The central portion of the organelle is referred to as the IMC body. This portion of the organelle contains proteins with a wide variety of functions such as maintaining parasite structure and hosting the actin-myosin motor that facilitates parasite gliding motility [23,24]. Finally, the base of the IMC contains a ring-shaped structure called the basal complex, which mediates bud constriction at the conclusion of cell division [25]. Directly underneath the IMC there is an array of 22 subpellicular microtubules (SPMTs) which originate in the apical complex of the parasite and extend approximately two-thirds of the way down the parasite body, providing further structural support [26].

One of the critical functions of the IMC is to act as a scaffold for developing daughter cells during parasite replication. While the exact mechanism varies across the phylum, many apicomplexan parasites replicate using a unique process of internal budding in which two or more daughter buds are formed within the cytoplasm of a maternal parasite [27,28]. The human-infecting form of *T. gondii* replicates asexually using a form of internal budding called

endodyogeny, in which two daughter buds are formed. Endodyogeny occurs in a series of tightly regulated steps, and parasites within a single vacuole typically replicate synchronously [27,29]. The first step of endodyogeny, bud initiation, occurs immediately after centrosome duplication. During bud initiation, the daughter cell scaffold assembles on top of the centrosome as early IMC proteins are recruited and the tubulin-based conoid and SPMTs are formed [19,30–33]. As endodyogeny continues, the daughter buds elongate, driven by polymerization of the SPMTs [34]. As buds approach maturation, the basal complex of the nascent daughter buds constricts and the maternal IMC is degraded [25,35]. Finally, the two daughter buds adopt the maternal plasma membrane and emerge as separate cells. Throughout this process, IMC proteins are synthesized and recruited to the daughter cell scaffold sequentially in a "just in time" manner, with some proteins appearing as early as bud initiation and others only appearing after bud maturation is complete [19,36,37]. Interestingly, some IMC proteins have been shown to localize only to the developing daughter buds and are removed during bud maturation and emergence [14,19,38–40].

Until recent years, it was unclear which IMC proteins served as the essential foundation of the organelle during endodyogeny. Several of the alveolins are thought to be essential, but most of these proteins recruit after budding has already been initiated [19]. They are also maintained in mature parasites where they play critical roles in maintaining cellular structure. Other proteins such as IMC15, IMC29, FBXO1, and MORN1 have been shown to recruit during bud initiation and play important roles in endodyogeny, but all four of these can be genetically disrupted [19,38,41,42]. We recently identified two proteins, IMC32 and IMC43, that form a complex in the early daughter cell scaffold and are essential for the stable assembly of the IMC during endodyogeny [39,40]. We demonstrated that a domain towards the C-terminus of IMC43 binds directly to the C-terminal coiled-coil (CC) domains of IMC32. While IMC32 initially recruits independently of IMC43 during bud initiation, IMC43 binding was found to be essential for the maintenance of IMC32's localization during the middle and late stages of endodyogeny. Since loss of either of these proteins results in a lethal inability to stably assemble the daughter IMC, we termed this the IMC43-IMC32 essential daughter bud assembly complex.

In this study, we set out to identify additional components of the IMC43-IMC32 essential daughter bud assembly complex. Using IMC32 as bait in both proximity labeling and yeast two-hybrid (Y2H) screens, we identify the essential early daughter IMC protein BCC0 as the third component of this complex and subsequently show that BCC0 localization is dependent on both IMC32 and IMC43. Deletion analyses and pairwise Y2H assays reveal that a domain towards the N-terminus of BCC0 is essential for both the protein's localization and function and for binding to its partner, IMC32. This work functionally connects the only three known essential early daughter IMC proteins and expands our understanding of how the early daughter IMC is assembled during endodyogeny.

## Results

### Identification of candidate IMC32 binding partners

To identify additional components of the IMC43-IMC32 essential daughter bud assembly complex, we used IMC32 as the bait in both a TurboID proximity labeling screen and a Y2H screen [43–45]. For the TurboID experiment, we fused sequences encoding a TurboID biotin ligase along with a 3xHA epitope tag to the C-terminus of the IMC32 endogenous locus (Fig 1A). Immunofluorescence assay (IFA) confirmed that the IMC32<sup>TurboID</sup> fusion protein localized normally to the daughter IMC (Fig 1B). After four hours of treatment with biotin, daughter buds were robustly biotinylated, confirming that the biotin ligase was enzymatically active.

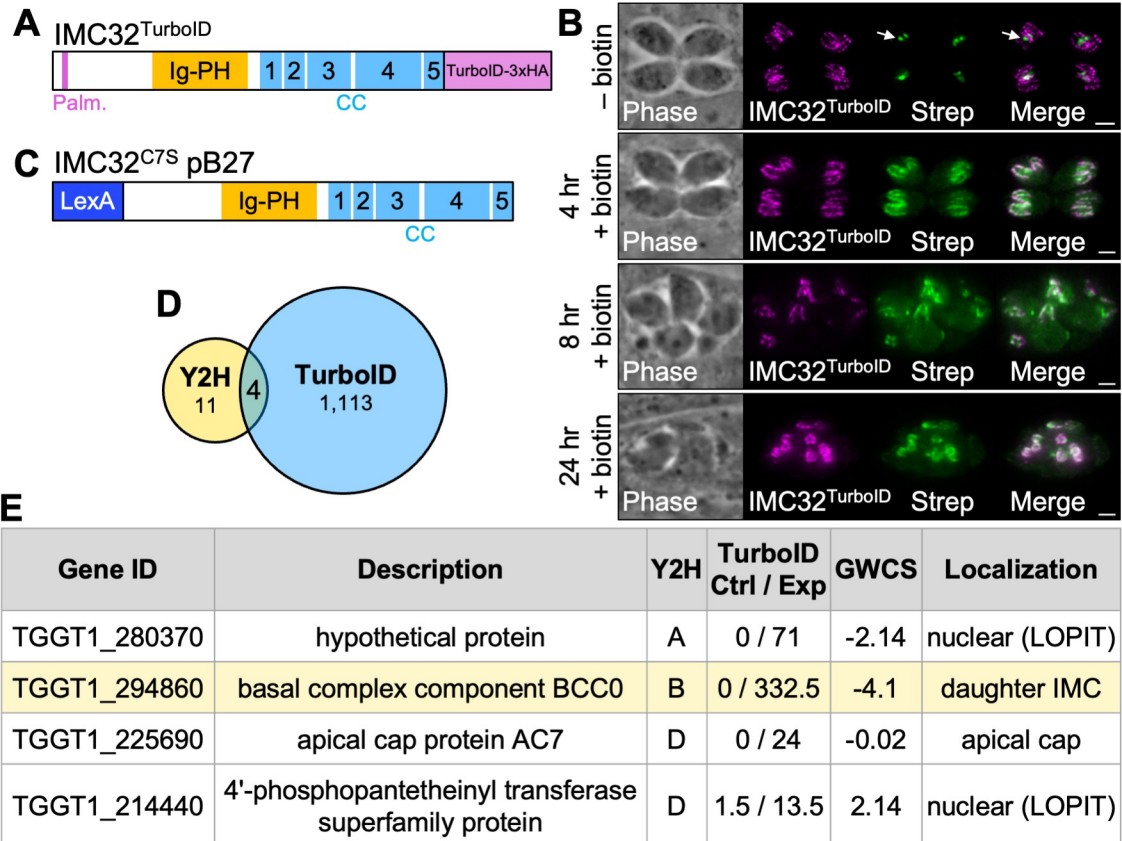

**Fig 1. TurboID and Y2H screens identify BCC0 as a candidate IMC32 binding partner.** A) Diagram of IMC32<sup>TurboID</sup> showing its predicted palmitoylation site (Palm.), Ig-PH domain, and five coiled-coil (CC) domains. A 3xHA-tagged TurboID biotin ligase was added to the C-terminus of the protein. B) IFAs of IMC32<sup>TurboID</sup> parasites after 0, 4, 8, or 24 hours of biotin treatment. Arrow points to the endogenously biotinylated apicoplast in the–biotin control. Morphological defects are visible at the 8 and 24-hour time points. Magenta = anti-HA detecting IMC32<sup>TurboID</sup>, Green = streptavidin. Scale bars = 2 μm. C) Diagram of the IMC32<sup>C7S</sup> N-terminal LexA fusion protein used for Y2H screening. D) Venn diagram comparing the genes identified in the TurboID and Y2H screens. All Y2H hits that were not out-of-frame or antisense were included. TurboID results were filtered to include only genes that were at least two-fold enriched with a difference of >5 spectral counts when comparing IMC32<sup>TurboID</sup> to control. Four genes were identified in both experiments after filtering results as described. E) Table showing the four genes that were identified in both the TurboID and Y2H screens. "Y2H" column indicates the confidence score assigned in the Y2H screen (A = very high confidence, B = high confidence, D = moderate confidence). "TurboID Ctrl / Exp" column indicates the average spectral count for each gene in the control and IMC32<sup>TurboID</sup> mass spectrometry results. "GWCS" refers to the phenotype score assigned to each gene in a genome-wide CRISPR/Cas9 screen [65]. "Localization" column reports the known localization of each protein [14,38,46]. Localizations followed by "(LOPIT)" indicate predicted localizations based on hyperplexed localization of organelle proteins by isotope tagging (hyperLOPIT) [83].

However, after eight hours of treatment with biotin, IMC32<sup>TurboID</sup> parasites exhibited morphological defects which became more severe by 24 hours. To avoid issues with toxicity, we treated parasites with biotin for five hours before harvesting and performing streptavidin affinity chromatography and subsequent mass spectrometry analysis (S1 Table). Y2H screening was performed by Hybrigenics Services using full-length IMC32 with the predicted palmitoylation site at position 7 mutated to serine. This protein was cloned into the pB27 bait vector as a LexA N-terminal fusion (IMC32<sup>C7S</sup> pB27) and screened against the *T. gondii* RH strain cDNA library to identify direct protein-protein interactions (Fig 1C and S2 Table). The results of the IMC32 TurboID and Y2H screens were compared, resulting in the identification of four candidate binding partners that were found in both approaches: a hypothetical protein encoded by the gene ID TGGT1_280370, the essential early daughter IMC protein BCC0, the

apical cap protein AC7, and a 4'-phosphopantetheinyl transferase superfamily protein (Fig 1D and 1E). BCC0 was a high confidence hit in the Y2H screen and the sixth most enriched protein in the TurboID screen. In addition, it has previously been shown to localize to daughter buds in a similar pattern to IMC32 and play an essential role in endodyogeny [38,46]. We therefore selected it for further study.

## BCC0 depends on IMC32 and IMC43 for its localization to the daughter IMC

BCC0 is a large protein composed of 2,457 amino acids (Fig 2A) [47]. The protein is predicted to contain a myristoylation site, three palmitoylation sites, and a coiled-coil (CC) domain at the N-terminus of the protein [48–51]. Our Y2H screen determined that residues 570–899 of BCC0 were involved in the interaction with the bait protein IMC32 (S2 Table). To determine the extent to which BCC0 and IMC32 colocalize, we endogenously tagged BCC0 with a spaghetti monster (sm) OLLAS epitope tag in an IMC32$^{\text{2xStrep3xTy}}$ parent strain. IFA showed that the two proteins exhibited near-perfect overlap during both bud initiation, when both proteins appear as five distinct puncta arranged symmetrically, and mid-budding, when both proteins appear as a series of five discontinuous longitudinal stripes along the body of the daughter IMC (Fig 2B and 2C). To determine whether BCC0's localization was dependent on either of the other two proteins in the essential daughter bud assembly complex (Fig 2D), we endogenously tagged BCC0 in both the IMC32$^{\text{AID}}$ and IMC43$^{\text{AID}}$ strains [40,52,53]. After 24 hours of indoleacetic acid (IAA) treatment, BCC0 became severely mislocalized in both IMC32-depleted and IMC43-depleted parasites (Fig 2E and 2F). Since IMC32's localization is dependent on binding to IMC43, this suggests that the complex assembles hierarchically.

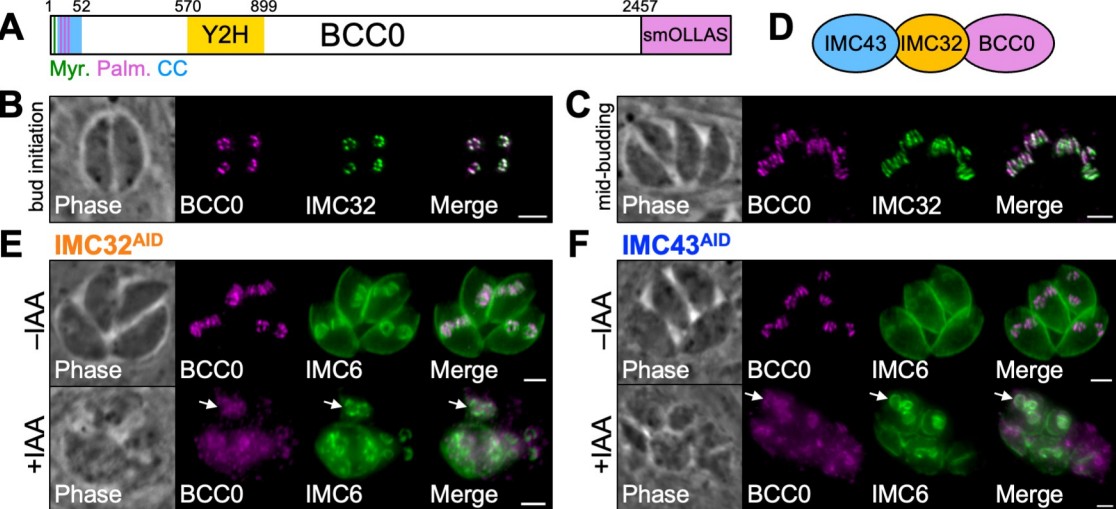

**Fig 2. BCC0 requires both IMC32 and IMC43 for its localization to the daughter IMC.** A) Diagram of BCC0 showing its predicted myristoylation site at residue 2 (Myr.), predicted palmitoylation sites at residues 10, 11, and 17 (Palm.), predicted coiled-coil domain at residues 9–52 (CC), and IMC32-binding region at residues 570–899 identified by Y2H screening (Y2H). An smOLLAS tag was added to the C-terminus of the protein. B-C) IFAs showing that BCC0 and IMC32 colocalize closely in the daughter IMC during both bud initiation and mid-budding. Magenta = anti-OLLAS detecting BCC0$^{\text{smOLLAS}}$, Green = anti-Ty detecting IMC32$^{\text{2xStrep3xTy}}$. D) Diagram of the essential daughter bud assembly complex composed of IMC43, IMC32, and BCC0. E) IFA of IMC32$^{\text{AID}}$ parasites grown -/+ IAA for 24 hours showing that depletion of IMC32 causes BCC0 to mislocalize to the cytoplasm but remain slightly enriched at daughter buds (arrows). Magenta = anti-OLLAS detecting BCC0$^{\text{smOLLAS}}$, Green = anti-IMC6. F) IFA of IMC43$^{\text{AID}}$ parasites grown -/+ IAA for 24 hours showing that depletion of IMC43 causes BCC0 to mislocalize to the cytoplasm but remain slightly enriched at daughter buds (arrows). Magenta = anti-OLLAS detecting BCC0$^{\text{smOLLAS}}$, Green = anti-IMC6. Scale bars = 2 μm.

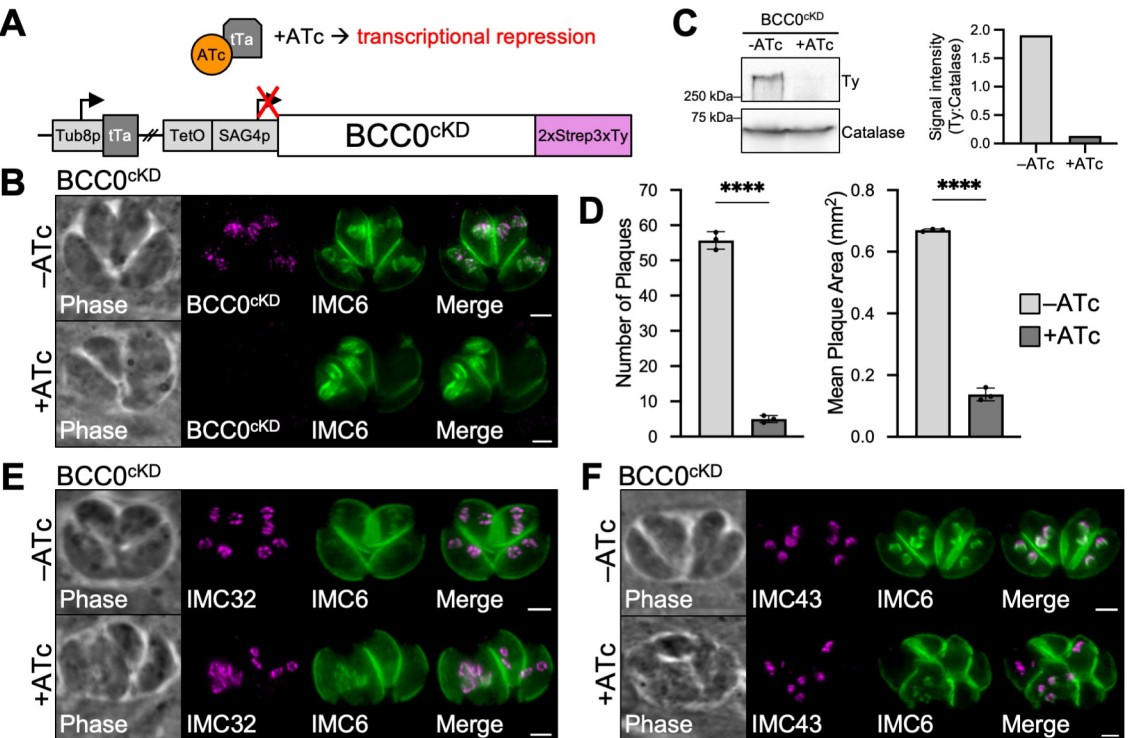

**Fig 3. BCC0 knockdown does not affect the localization of IMC32 or IMC43.** A) Diagram depicting the ATc-regulatable transcriptional repression system used for BCC0 knockdown. The endogenous promoter for BCC0 was replaced with a TetO7-SAG4p minimal promoter in a BCC0[2xStrep3xTy] strain expressing the Tati transactivator. B) IFA of BCC0[cKD] parasites grown -/+ ATc for 24 hours showing that knockdown of BCC0 results in severe morphological defects. Magenta = anti-Ty detecting BCC0[cKD], Green = anti-IMC6. C) Western blot and quantification showing that BCC0[cKD] is 93% depleted after 24 hours of ATc treatment. D) Plaque assays show that both plaque number and size are significantly reduced when BCC0 is knocked down. Statistical significance was determined using a two-tailed t-test (****, P < 0.0001). E) IFA of BCC0[cKD] parasites grown -/+ ATc for 24 hours showing that knockdown of BCC0 does not affect IMC32 localization. Magenta = anti-Myc detecting IMC32[3xMyc], Green = anti-IMC6. F) IFA of BCC0[cKD] parasites grown -/+ ATc for 24 hours showing that knockdown of BCC0 does not affect IMC43 localization. Magenta = anti-HA detecting IMC43[smHA], Green = anti-IMC6. Scale bars = 2 μm.

We next wanted to determine how loss of BCC0 would affect the localization of IMC32 and IMC43. We initially tried to use the auxin-inducible degron (AID) system, but endogenously tagging BCC0 with either the mAID[3xHA] or the mIAA7[3xHA] degron resulted in insufficient knockdown (S1 Fig) [54]. Thus, we utilized a transcriptional repression system to study BCC0 [55,56]. To do this, we endogenously tagged BCC0 with a 2xStrep3xTy epitope tag in an RHΔ*ku80*-Tati-HXGPRT parent strain, then replaced the endogenous promoter with a TetO7-SAG4 minimal promoter (BCC0[cKD]) (Fig 3A). Treatment with anhydrotetracycline (ATc) for 24 hours caused the protein to be knocked down to an undetectable level by IFA and resulted in severe morphological defects, as expected based on published data (Fig 3B) [46]. Western blot confirmed that 24 hours of ATc treatment resulted in a 93% reduction in the level of BCC0 expression (Fig 3C). Plaque assays demonstrated that knockdown of BCC0 led to a severe 92% reduction in plaque number and an 80% reduction in plaque size (Fig 3D). These few small plaques are likely formed due to the incomplete knockdown observed by western blot (Fig 3C). To determine the effect of BCC0 knockdown on IMC32 and IMC43, we endogenously tagged IMC32 with a 3xMyc tag and IMC43 with an smHA tag in the BCC0[cKD] strain. IFA demonstrated that despite the severe morphological defects observed in these parasites, IMC32 and IMC43 localization remained unaffected (Fig 3E and 3F).

## Residues 701–877 are essential for BCC0 localization and function

Since BCC0 is a large protein with no identifiable functional domains or homology to known proteins, we next wanted to determine which regions of the protein are important for its localization and function. To do this, we created a series of 12 smHA-tagged constructs for functional complementation studies (Fig 4A). These included the full-length protein (BCC0$^{WTc}$) as a control and 11 mutant proteins each containing a deletion of approximately 150–300 amino acids. Regions for the deletions were chosen based on conservation with *N. caninum* and predicted secondary structure (S2 Fig) [57]. Each construct was driven by the endogenous promoter for BCC0 and contained flanking regions for integration at the UPRT locus in a BCC0$^{smOLLAS}$ strain (Fig 4B). Localization was assessed by co-staining for the smHA-tagged construct and wild-type BCC0$^{smOLLAS}$. We then attempted to disrupt the endogenous BCC0 locus in each of these strains. Successful knockouts were assessed by IFA and plaque assay to determine whether each deletion could fully complement the function of wild-type BCC0.

BCC0$^{WTc}$ and 10 of the deletion constructs colocalized perfectly with endogenous BCC0$^{smOLLAS}$ (Figs 4C and S3). However, BCC0$^{\Delta701-877}$ became severely mislocalized with most of the protein being mistargeted to the cytoplasm, although there was a slight enrichment at daughter buds (Fig 4D). Attempts to disrupt the endogenous BCC0 locus in the BCC0$^{smOLLAS}$ + BCC0$^{\Delta701-877}$ strain were unsuccessful, indicating that this region of the protein is likely required for its function. We were able to successfully disrupt the BCC0 locus in the BCC0$^{WTc}$ strain and all other deletion strains, which was confirmed by both PCR verification and IFA showing loss of the BCC0$^{smOLLAS}$ signal (S4 Fig). No obvious morphological defects were observed for any of these strains, although the $\Delta bcc0$ + BCC0$^{\Delta170-375}$ strain often exhibited desynchronized endodyogeny, suggesting possible dysregulation (Fig 4E–4O). Plaque assays demonstrated that BCC0$^{WTc}$ and all deletions except for BCC0$^{\Delta170-375}$ were sufficient to maintain normal growth (Fig 4P). The $\Delta bcc0$ + BCC0$^{\Delta170-375}$ strain exhibited a significant 32% reduction in plaque size, indicating this region likely plays a minor role in the function of BCC0. The $\Delta2-169$ and $\Delta376-569$ deletions, which flank this region, exhibited modest but nonsignificant reductions in plaque size (17% and 12%, respectively).

Since we were unable to disrupt the endogenous BCC0 locus in the BCC0$^{smOLLAS}$ + BCC0$^{\Delta701-877}$ strain, we integrated the BCC0$^{WTc}$ and BCC0$^{\Delta701-877}$ constructs at the UPRT locus in the BCC0$^{cKD}$ strain and used IFA and plaque assays to assess whether each construct could rescue the defects observed upon BCC0 knockdown (Fig 5A and 5B). While the BCC0$^{WTc}$ construct was able to fully rescue the morphological defects observed by IFA, the BCC0$^{\Delta701-877}$ construct could not (Fig 5C and 5D). Knockdown of BCC0$^{cKD}$ resulted in an 83% reduction in plaque efficiency and a 62% reduction in plaque size (Fig 5E and 5F). Both of these defects were slightly more modest than what we previously observed, suggesting that the leaky expression of BCC0$^{cKD}$ may have increased over time (Fig 3C and 3D). The BCC0$^{WTc}$ construct restored plaque size to approximately 80% the size of plaques formed by untreated parasites, as expected based on ATc toxicity in wild-type parasites (Figs 5E and S5) [40]. The BCC0$^{\Delta701-877}$ construct completely failed to rescue the defect in both plaque efficiency and plaque size caused by knockdown of BCC0$^{cKD}$, confirming that this region is essential for both the localization and function of BCC0 (Fig 5E and 5F).

## Residues 1–899 are sufficient for BCC0 function

Since only residues 701–877 are necessary for BCC0 function, we wondered how much of the protein was sufficient for its function. To determine this, we began by making C-terminal truncations of the protein at its endogenous locus. We found that BCC0 was able to be truncated at residue 1519 (BCC0$^{1-1519}$) without having any effect on the protein's localization

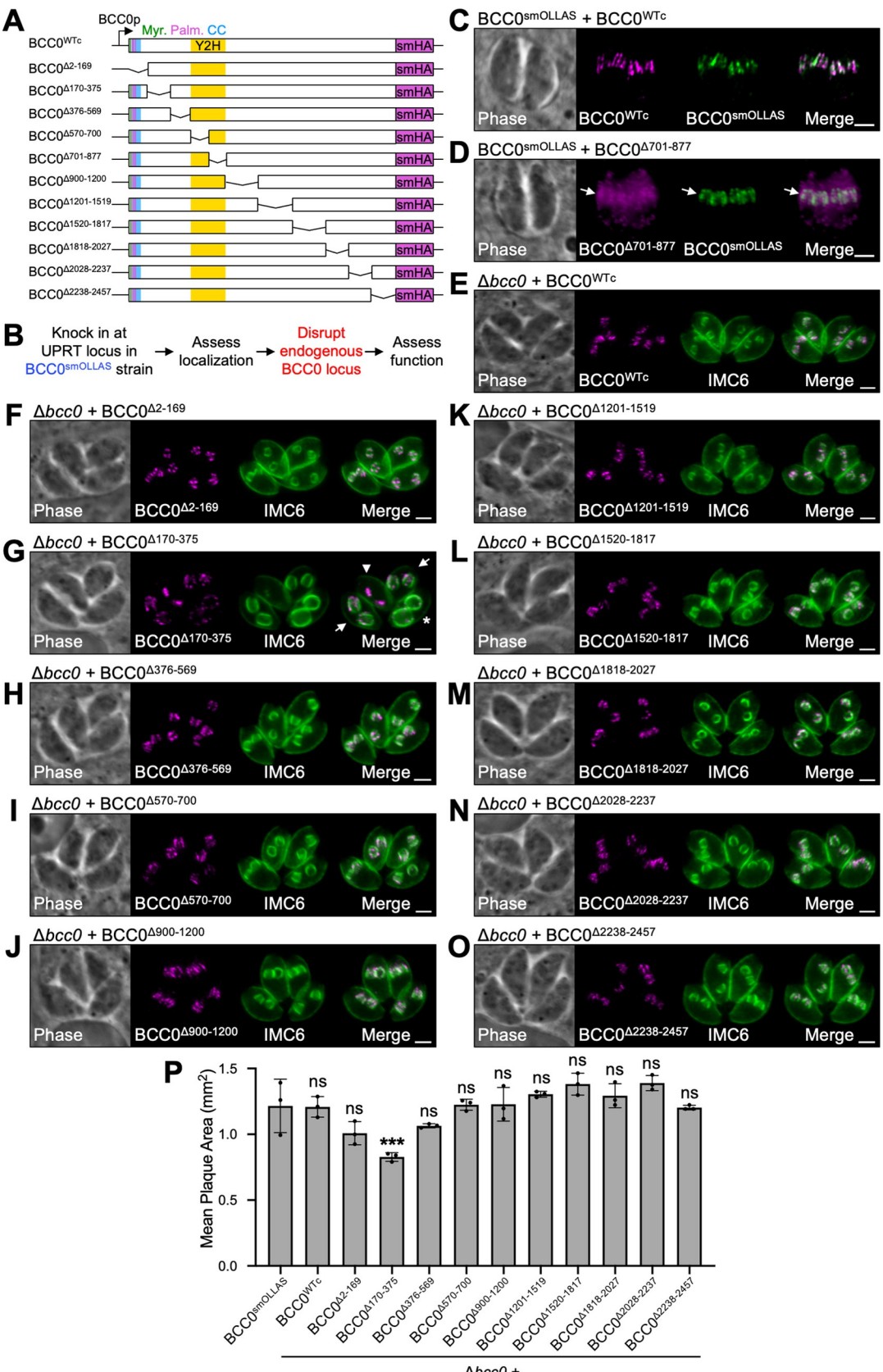

**Fig 4. BCC0 residues 701–877 are required for localization and residues 170–375 have a minor impact on function.** A) Diagram showing the 12 constructs used for functional complementation of BCC0. All 12 constructs were driven by the endogenous promoter for BCC0 and include a C-terminal smHA tag. "Myr." = predicted myristoylation site. "Palm." = predicted palmitoylation sites. "Y2H" = IMC32-binding region identified by Y2H screen. B) Workflow used to assess the effect of each region of BCC0 on the protein's localization and function. C) IFA of BCC0$^{smOLLAS}$ + BCC0$^{WTc}$ parasites showing that BCC0$^{WTc}$ localizes normally. Magenta = anti-HA detecting BCC0$^{WTc}$, Green = anti-OLLAS detecting BCC0$^{smOLLAS}$. D) IFA of BCC0$^{smOLLAS}$ + BCC0$^{\Delta701-877}$ parasites showing that BCC0$^{\Delta701-877}$ severely mislocalizes to the cytoplasm and is slightly enriched at daughter buds (arrows). Magenta = anti-HA detecting BCC0$^{\Delta701-877}$, Green = anti-OLLAS detecting endogenous BCC0$^{smOLLAS}$. E) IFA of $\Delta bcc0$ + BCC0$^{WTc}$ parasites which exhibit no obvious morphological defects. Magenta = anti-HA detecting BCC0$^{WTc}$, Green = anti-IMC6. F-O) IFAs for all deletion strains in which endogenous BCC0$^{smOLLAS}$ was successfully disrupted. BCC0$^{\Delta701-877}$ is not shown because endogenous BCC0$^{smOLLAS}$ was unable to be disrupted in this background. Panel G shows $\Delta bcc0$ + BCC0$^{\Delta170-375}$ parasites undergoing desynchronized endodyogeny. An arrowhead indicates a parasite at the early budding stage, arrows indicate parasites at the mid-budding stage, and an asterisk indicates a parasite at the late budding stage. Magenta = anti-HA detecting BCC0 deletions, Green = anti-IMC6. P) Plaque assays for all deletion strains in which endogenous BCC0$^{smOLLAS}$ was successfully disrupted. BCC0$^{\Delta701-877}$ is not shown because endogenous BCC0$^{smOLLAS}$ was unable to be disrupted in this background. Only the $\Delta bcc0$ + BCC0$^{\Delta170-375}$ strain exhibited a significant reduction in plaque size. Statistical significance was determined by one-way ANOVA comparing each strain to BCC0$^{smOLLAS}$ (\*\*\*, P < 0.001; ns = not significant). Scale bars for all IFAs = 2 μm.

(Fig 6A). Truncation at residue 899 (BCC0$^{1-899}$) resulted in partial mislocalization of the protein, which was most apparent in the later stages of endodyogeny (Fig 6B). Despite this partial mislocalization, this truncated protein did not exhibit a fitness defect (Fig 6C). Since any further truncation would disrupt the essential domain within residues 701–877, we next asked whether additional deletions could be made at the N-terminus of the protein. To determine this, we designed four constructs containing residues 170–899, 376–899, 570–899, and 701–899 and integrated them at the UPRT locus in a wild-type BCC0$^{smOLLAS}$ strain (Fig 6D). As we had done previously for our initial deletion series, we assessed the localization of each of these mutant proteins and then disrupted the endogenous BCC0 locus to assess its function (Fig 6E). All four mutant proteins severely mislocalized to the cytoplasm (S6A–S6D Fig). Surprisingly, despite this, we were able to successfully disrupt the endogenous BCC0 locus in all four of these strains, which was confirmed by IFA and PCR (Figs 6F–6I and S6E). Each of the four mutant strains exhibited obvious morphological and replication defects and a severe reduction in plaque size (Fig 6F–6J). Thus, while just the essential domain of BCC0 (BCC0$^{701-899}$) is sufficient for parasite survival, only BCC0$^{1-899}$ is sufficient for full function of the protein.

## BCC0's essential domain binds to IMC32's coiled-coil domains

Finally, we wanted to use pairwise Y2H assays to determine which region of IMC32 is involved in binding to BCC0 and to determine whether there are any direct interactions between BCC0 and IMC43. We divided BCC0 into four fragments: BCC0$^{3-569}$, BCC0$^{570-899}$, BCC0$^{900-1817}$, and BCC0$^{1818-2457}$ (Fig 7A). For IMC32 and IMC43, we used the same fragments that we had previously used in our investigation of IMC32-IMC43 binding interactions [40]. Each fragment was cloned into either a pB27 (LexA) bait vector or a pP6 (GAL4 activation domain) prey vector. We then co-transformed all 12 possible combinations of the IMC32 and BCC0 bait and prey vectors into yeast and tested each strain's ability to grow on permissive and restrictive media (Fig 7B). Four of the combinations could not be tested because both fragments were auto-activating in the context of the pB27 vector. Of the remaining eight strains, only the strain expressing BCC0$^{570-899}$ and IMC32$^{Cterm}$ was able to grow on restrictive media, indicating that BCC0's essential domain binds to IMC32 at its C-terminal coiled-coil domains. To determine whether BCC0 directly binds to IMC43, we performed the same experiments with all 12 combinations of IMC43 and BCC0 bait and prey vectors. All 12 strains grew robustly on permissive media and were not auto-activating, but none grew on permissive media, indicating that there are no direct interactions between these two proteins (Fig 7C).

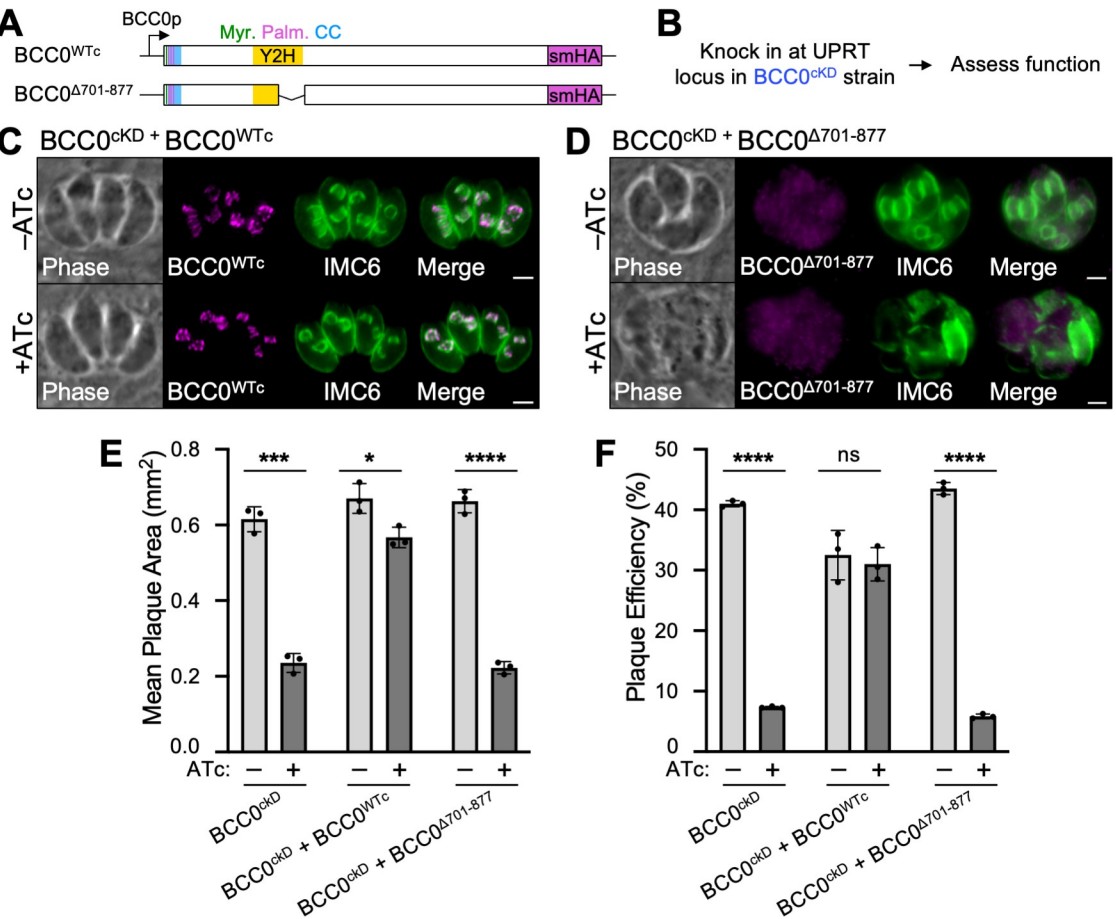

**Fig 5. BCC0 residues 701–877 are essential for function.** A) Diagram of the two constructs used for functional complementation of BCC0$^{cKD}$. Both are driven by the endogenous promoter for BCC0 and include a C-terminal smHA tag. "Myr." = predicted myristoylation site, "Palm." = predicted palmitoylation site, "CC" = predicted coiled-coil domain, "Y2H" = IMC32-binding domain identified by Y2H screen. B) Workflow used to assess how deleting residues 701–877 affects the function of BCC0 using an ATc-regulatable conditional knockdown strain (BCC0$^{cKD}$). C) IFA of BCC0$^{cKD}$ + BCC0$^{WTc}$ parasites grown for 24 hours -/+ ATc showing that BCC0$^{WTc}$ rescues the morphological defects caused by knockdown of BCC0. Magenta = anti-HA detecting BCC0$^{WTc}$, Green = anti-IMC6. D) IFA of BCC0$^{cKD}$ + BCC0$^{\Delta 701-877}$ parasites grown for 24 hours -/+ ATc showing that BCC0$^{\Delta 701-877}$ fails to rescue the morphological defects caused by knockdown of BCC0. Magenta = anti-HA detecting BCC0$^{\Delta 701-877}$, Green = anti-IMC6. E, F) Plaque assays show that BCC0$^{\Delta 701-877}$ fails to rescue the defect in both plaque efficiency (number of plaques formed divided by the number of parasites infected) and plaque size caused by knockdown of BCC0. BCC0$^{WTc}$ does not completely rescue the defect in plaque size due to ATc toxicity which is documented in S5 Fig and previous work [39]. Statistical significance was determined using multiple two-tailed t-tests (****, P < 0.0001; ***, P < 0.001; *, P < 0.05; ns = not significant). Scale bars for all IFAs = 2 μm.

## Discussion

In this study, we identified the essential daughter IMC protein BCC0 as a binding partner of IMC32 and a third component of the essential daughter bud assembly complex (Fig 8A). Using a combination of deletion constructs integrated at an exogenous locus and C-terminal truncations at the endogenous locus, we were able to interrogate how the loss of different regions of BCC0 impacts the protein's localization and function. One surprising result was that BCC0$^{\Delta 2-169}$, which lacks the predicted N-terminal acylation sites, was able to localize to the daughter IMC and function normally. Several IMC proteins such as IMC32, ISP1-4, GAP45, and HSP20 have been shown to depend on their predicted acylation sites for localization to the IMC [16,17,39,58,59]. However, MORN1 has a predicted palmitoylation site which has been shown to be dispensable for its localization to the basal complex. It is possible that

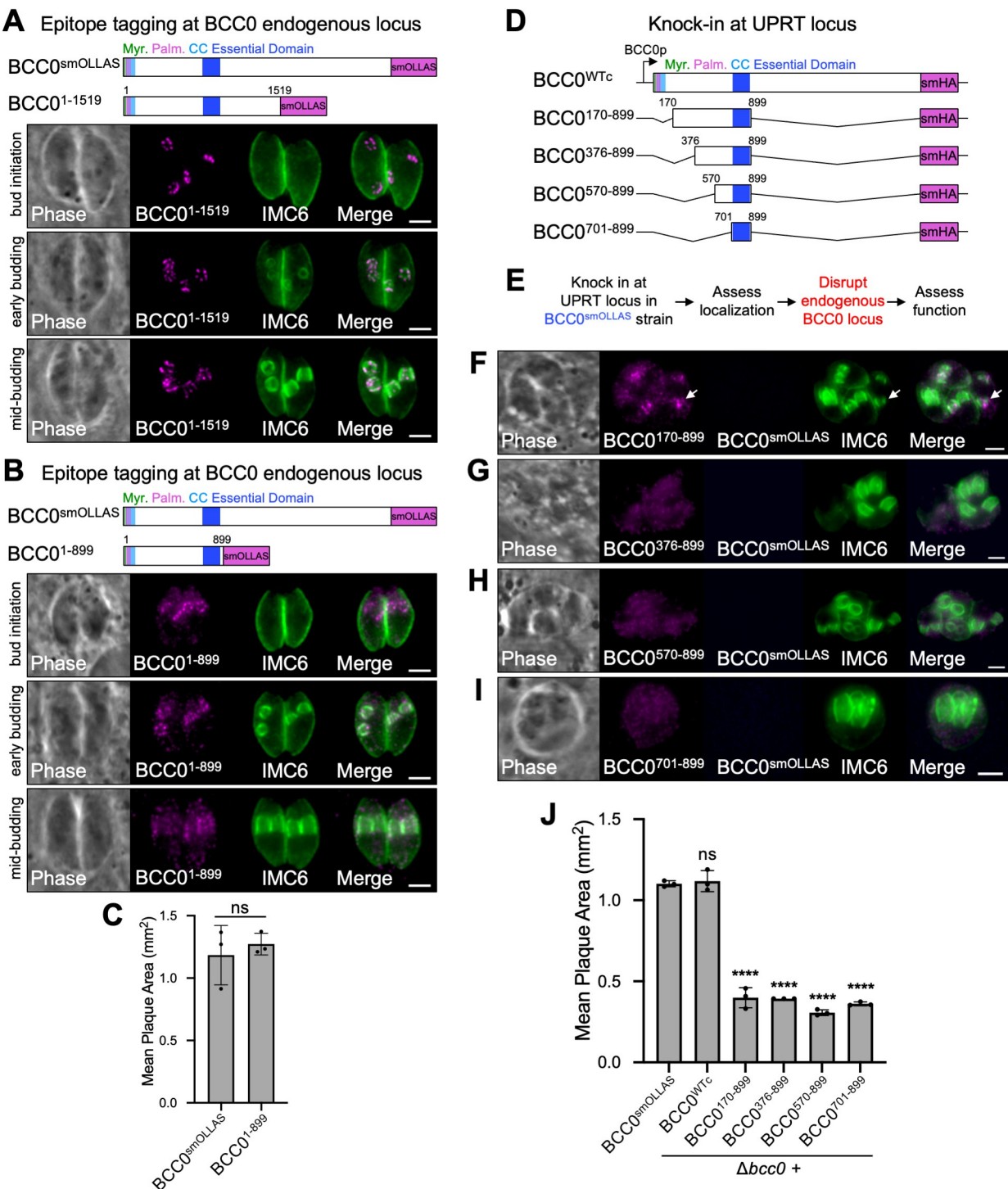

**Fig 6. BCC0 residues 1–899 are sufficient for function.** A) Diagram and IFAs showing that BCC0 can be truncated at residue 1519 without affecting localization of the protein during the bud initiation, early budding, or mid-budding stages of endodyogeny. For diagrams, "Myr." = predicted myristoylation site, "Palm." = predicted palmitoylation site, "CC" = coiled-coil domain. For IFAs, Magenta = anti-OLLAS detecting BCC0[1-1519], Green = anti-IMC6. B) Diagram and IFAs showing that truncation of BCC0 at residue 899 causes partial mislocalization of the protein which is most severe at the mid-budding stage of endodyogeny. Magenta = anti-OLLAS detecting BCC0[1-899], Green = anti-IMC6. C) Plaque assay comparing full-length BCC0[smOLLAS] with BCC0[1-899] shows that truncation of BCC0 at residue 899 does not affect function. Statistical significance was determined using a two-tailed t-test (ns = not significant). D) Diagram of the four deletion constructs used to determine the minimal region of BCC0 that is sufficient for protein function. All constructs are driven by the endogenous promoter for BCC0 and include a C-terminal smHA tag. E) Workflow used to analyze the localization and function of each of the constructs shown in D. F-I) IFAs of Δ*bcc0* parasites

complemented with each of the constructs shown in D. All four constructs severely mislocalize, and only BCC0[170-899] enriches at daughter buds (arrow). Magenta = anti-HA detecting BCC0 deletions, Blue = anti-OLLAS detecting endogenous BCC0[smOLLAS], Green = anti-IMC6. J) Plaque assays comparing the size of plaques formed by wild-type BCC0[smOLLAS] parasites with that of Δ*bcc0* parasites complemented with the constructs shown in D. Statistical significance was determined by one-way ANOVA comparing each strain to BCC0[smOLLAS] (****, P < 0.0001; ns = not significant). Scale bars for all IFAs = 2 μm.

these predicted sites on BCC0 are not actually acylated [42]. Alternatively, BCC0 could have bona fide acylation sites which only serve as secondary means of tethering to the IMC.

Our deletion series revealed that residues 701–877 of BCC0 are essential for both its localization and function. Since these residues are contained within the region which we found to be sufficient for IMC32 binding by pairwise Y2H assays and depletion of IMC32 also causes mislocalization of BCC0, this suggests BCC0 is primarily targeted to the daughter cell scaffold by binding to IMC32. We were surprised to find that BCC0 could be truncated down to residue 899 with only minor effects on the protein's localization and no impact on the protein's function. Even more surprisingly, expression of just the essential domain of BCC0, residues 701–877, was sufficient for parasite viability, although the protein was severely mislocalized and these parasites exhibited defects in growth and morphology. This suggests that a small

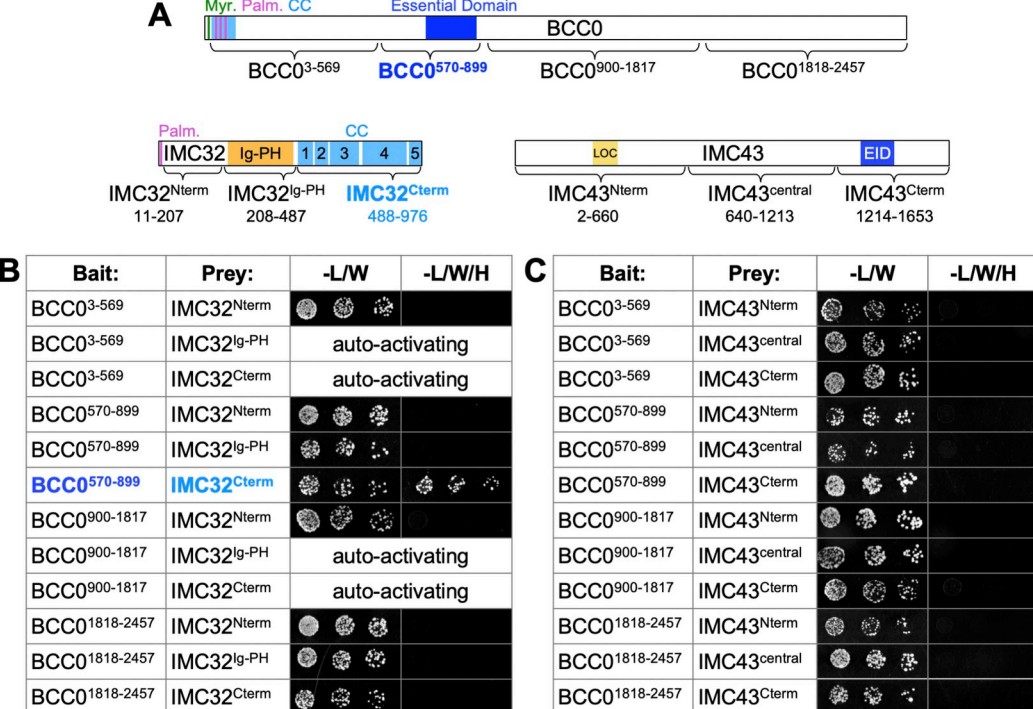

**Fig 7. BCC0 residues 570–899 bind to IMC32's C-terminal coiled-coil domains.** A) Diagram of the fragments of BCC0, IMC32, and IMC43 used for pairwise Y2H assays. Residues included in each fragment are indicated. "Myr." = predicted myristoylation site. "Palm." = predicted palmitoylation site. "CC" = coiled-coil domain. "LOC" = IMC43 localization domain. "EID" = IMC43 essential interaction domain. B) Pairwise Y2H assays testing for binding between fragments of BCC0 and IMC32. Pairs of fragments that were unable to be tested due to auto-activation of both fragments in the pB27 vector are indicated. Growth on permissive (-L/W) media indicates the presence of both the pB27 and pP6 bait and prey plasmids. Growth on restrictive (-L/W/H) media indicates binding between the indicated fragments of each protein. C) Pairwise Y2H assays testing for binding between fragments of BCC0 and IMC43.

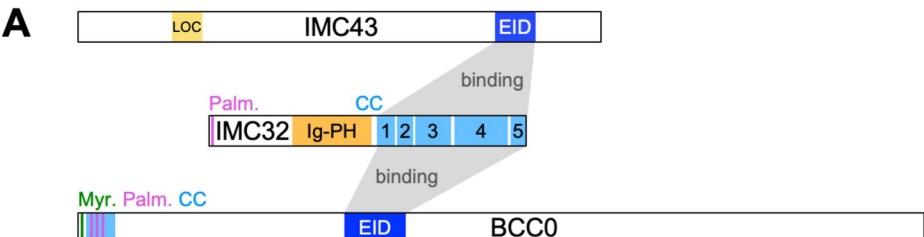

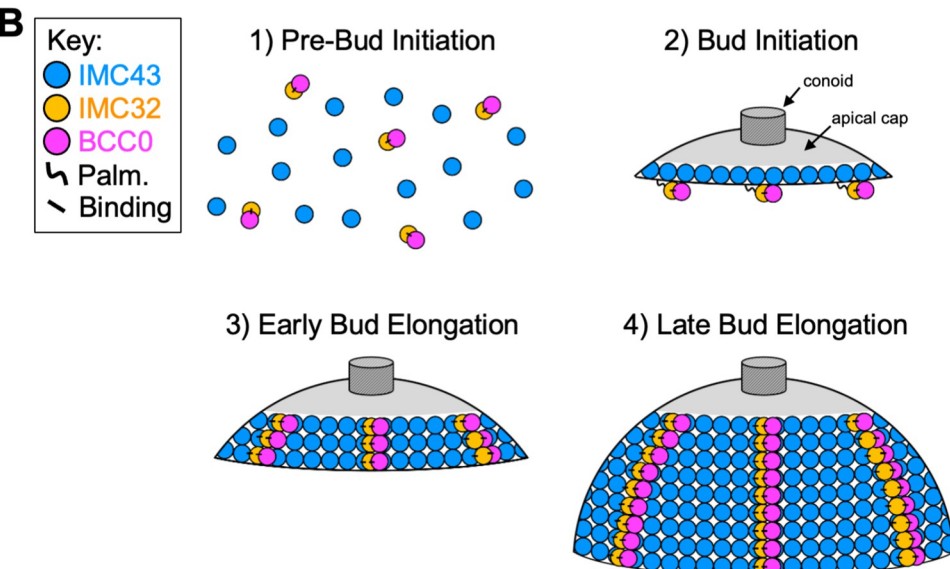

**Fig 8. Summary.** A) Diagram depicting interactions between the components of the essential IMC43-IMC32-BCC0 daughter bud assembly complex. Grey boxes indicate binding interactions between the essential interaction domains (EID) of IMC43 and BCC0 with IMC32's C-terminal coiled-coil domains. "Myr." = predicted myristoylation site. "Palm." = predicted palmitoylation site. "CC" = coiled-coil domain. B) Diagram summarizing the current model for how IMC43, IMC32, and BCC0 assemble onto the developing daughter cell scaffold during endodyogeny. Just before bud initiation (step 1) expression of IMC43, IMC32, and BCC0 increases. IMC32 and BCC0 bind to each other at this point. During bud initiation (step 2) IMC43 recruits to the early daughter cell scaffold independently. IMC32 is recruited to the membranes of the early daughter cell scaffold via palmitoylation. BCC0 remains bound to IMC32. During early bud elongation (step 3), IMC32 binds to IMC43, securely locking it into the daughter cell scaffold. BCC0 remains bound to IMC32. Short stripes of IMC32 and BCC0 can be seen at this point, which become more prominent as bud elongation continues (step 4).

percentage of BCC0$^{701-899}$ is able to localize to the daughter cell scaffold and provide sufficient function to support viability. Since $\Delta bcc0$ + BCC0$^{170-899}$ and $\Delta bcc0$ + BCC0$^{701-899}$ phenocopy each other, it could be possible that BCC0's primary function is carried out by the essential domain within residues 701–877, while a secondary function is carried out by residues 2–169. However, the fact that the BCC0$^{\Delta2-169}$ mutant was fully functional makes this less likely. Another possibility is that these proteins are more prone to misfolding due to the very large deletions on both the N- and C-termini of the protein. While interpretation of this portion of the data is difficult, the data from the original deletion series and endogenous C-terminal truncations clearly demonstrates that residues 701–877 are necessary and residues 1–899 are sufficient for the function of BCC0.

The data presented in this study and our previous work supports a hierarchical recruitment model for the essential IMC43-IMC32-BCC0 daughter bud assembly complex (Fig 8B). In our

initial studies of IMC32, we demonstrated that mutation of the predicted palmitoylation site causes IMC32 to mislocalize to the cytoplasm [39]. In our recent work in which we identified the IMC43-IMC32 complex, we also demonstrated that IMC32 localizes independently during bud initiation, but once bud elongation begins, IMC32 must bind to IMC43 in order to maintain its striped localization [40]. The data that we present in this study shows that BCC0 becomes mislocalized upon depletion of either IMC32 or IMC43 and that its localization is dependent on its IMC32-binding domain. Together, these data suggest that IMC32 and BCC0 bind to each other either prior to or during bud initiation and recruit to the early daughter cell scaffold via palmitoylation of IMC32. Then, once bud elongation begins, IMC32 binds to IMC43, locking the IMC32-BCC0 subcomplex into the daughter cell scaffold. It remains unclear how IMC43 is recruited to the developing daughter cell scaffold since it localizes normally in the absence of either IMC32 or BCC0 and does not contain any transmembrane domains or predicted acylation sites. It may bind to another IMC protein to tether it to either the alveoli or the cytoskeletal network of the developing daughter bud. Our previous IMC43 Y2H screen identified 28 additional candidate IMC43-binding partners. Future studies will be needed to explore these proteins. Additionally, the stoichiometry of the IMC43-IMC32-BCC0 complex is unclear. It is possible that the complex forms in a 1:1:1 manner, but it is also possible that one or more of the proteins is overrepresented in the complex, potentially allowing for multiple configurations. This would likely be resolved by structural studies of the complex.

Although our work has demonstrated the critical importance of the IMC43-IMC32-BCC0 daughter bud assembly complex and has yielded a model for how each component is recruited, the precise function of the complex remains unknown. Loss of any of the three complex components results in both severe defects in IMC morphology and dysregulation of endodyogeny [39,40,46]. Engelberg et al. additionally observed that loss of BCC0 led to subtle defects in the formation of the basal complex and apical annuli on developing daughter buds, although neither structure was fully disrupted [46]. Our previous work demonstrated that basal complex formation is unaffected by loss of IMC43, so it is possible that the IMC32-BCC0 subcomplex plays a role in recruiting MORN1 to the basal complex during bud initiation, before IMC43 joins the complex. This hypothesis is further supported by the different localizations of each protein during bud initiation. At this stage, both IMC32 and BCC0 are present in five distinct puncta which Engelberg et al. showed lie directly on top of the developing basal complex [39,46]. IMC43, on the other hand, appears as a slightly smaller continuous ring that lies slightly apical to the IMC32-BCC0 puncta [40]. Recent work has demonstrated that during bud initiation, the 22 subpellicular microtubules originally appear as five bundles of microtubules in a 4 + 4 + 4 + 4 + 6 configuration [30,60]. The fact that both the IMC32-BCC0 subcomplex and the nascent SPMTs exhibit the same 5-fold symmetry strongly suggests a functional relationship between these structures. We previously demonstrated that loss of IMC32 does not prevent formation of the conoid or SPMTs [39], but it could be possible that the five bundles of SPMTs act as a foundation for the formation of the IMC32-BCC0 puncta. In future studies it will be intriguing to explore how disruption of the SPMTs at this stage affects localization of the IMC32-BCC0 subcomplex using higher resolution techniques such as ultrastructure expansion microscopy.

Another outstanding question is how the essential daughter bud assembly complex is regulated. Our model suggests that IMC43 does not bind to IMC32 until after bud elongation has begun. How the timing of this binding event is controlled remains unknown. IMC43, IMC32, and BCC0 all have multiple phosphorylation sites which could represent possible means of regulation [61]. Our IMC43 Y2H screen identified three kinases (SRPK, CDPK7, and Ark3) and one phosphatase (PPKL) as candidate binding partners. CDPK7, Ark3, and PPKL have all been shown to play important roles in *T. gondii* endodyogeny [62–64]. SRPK has not been

studied, but its phenotype score suggests it may be essential [65]. Exploring how the activity of these enzymes affects the localization or function of IMC43 and its partners will be an interesting topic of future studies. Identification of additional complex components or regulatory proteins could be achieved by performing TurboID proximity labeling and Y2H screening using BCC0 as bait. BCC0's large size may make Y2H screening challenging due to toxicity, which we observed for IMC43. However, use of an inducible bait vector circumvented this problem for IMC43 and is likely to be successful for BCC0 as well [40]. Alternatively, since residues 1–899 are sufficient for the function of BCC0, it may be more practical to use the truncated protein as bait in a Y2H screen.

Together, the data reported in this study identifies the essential daughter IMC protein BCC0 as the third component of the IMC43-IMC32-BCC0 daughter bud assembly complex and presents a model for how the complex is formed. The functional complementation studies of BCC0 also demonstrate that despite its large size, much of the protein is dispensable and its critical functions are carried out by an essential interaction domain, similar to what we previously observed for IMC43 [40]. As this complex is foundational for the construction of *T. gondii* daughter buds during endodyogeny, a parasite-specific biological function, deepening our understanding of the function and regulation of this complex is likely to facilitate the identification of novel therapeutic targets for this important pathogen.

## Materials and methods

### *T. gondii* and host cell culture

Parental *T. gondii* RHΔ*hxgprt*Δ*ku80* (wild-type) and subsequent strains were grown on confluent monolayers of human foreskin fibroblasts (HFFs) (BJ, ATCC, Manassas, VA) at 37°C and 5% $CO_2$ in Dulbecco's Modified Eagle Medium (DMEM) supplemented with 5% fetal bovine serum (Gibco), 5% Cosmic calf serum (Hyclone), and 1x penicillin-streptomycin-L-glutamine (Gibco). Constructs containing selectable markers were selected using 1 μM pyrimethamine (dihydrofolate reductase-thymidylate synthase [DHFR-TS]), 50 μg/mL mycophenolic acid-xanthine (HXGPRT), or 40 μM chloramphenicol (CAT) [66–68]. Homologous recombination to the UPRT locus was negatively selected using 5 μM 5-fluorodeoxyuridine (FUDR) [69]. Knockdown of AID strains was performed by treating parasites with 0.5 mM IAA (Millipore Sigma, I2886). Knockdown of the BCC0^cKD strain was performed by treating parasites with 1 μg/mL ATc (Millipore Sigma, 37919).

### Antibodies

The HA epitope was detected with mouse monoclonal antibody (mAb) HA.11 (BioLegend; 901515). The Ty1 epitope was detected with mouse mAb BB2 [70]. The c-Myc epitope was detected with mouse mAb 9E10 [71]. The OLLAS epitope was detected with rat mAb anti-OLLAS [72]. *Toxoplasma*-specific antibodies include rabbit pAb anti-IMC6 [20] and rabbit anti-Catalase [73].

### Endogenous epitope tagging and knockout

For C-terminal endogenous tagging, a pU6-Universal plasmid containing a protospacer against the 3′ untranslated region (UTR) of the target protein approximately 100 bp downstream of the stop codon was generated, as described previously [74,75]. A homology-directed repair (HDR) template was PCR amplified from the Δ*ku80*-dependent LIC vector pmAID3xHA.LIC-HXGPRT, pmAID3xTy.LIC-HXGPRT, p3xHA.LIC-DFHR, p3xMyc.LIC-DFHR, p2xStrep3xTy.LIC-CAT, p2xStrep3xTy.LIC-HXGPRT, psmOLLAS.LIC-DFHR,

psmHA.LIC-DHFR, or pTurboID3xHA.LIC-DHFR, all of which include the epitope tag, 3′ UTR, and a selection cassette [76]. The 60-bp primers include 40 bp of homology immediately upstream of the stop codon or 40 bp of homology within the 3′ UTR downstream of the CRISPR/Cas9 cut site. For C-terminal truncations at the endogenous locus, the forward primer for amplifying the HDR template was designed using 40 bp of homology within the gene's coding sequence. Primers that were used for the pU6-Universal plasmid as well as the HDR template are listed in S3 Table (primers P1-P14).

For knockout of BCC0, the protospacer was designed to target an intron within the BCC0 locus, ligated into the pU6-Universal plasmid, and prepared similarly to the endogenous tagging constructs. The HDR template was PCR amplified from a pJET vector containing the HXGPRT drug marker driven by the NcGRA7 promoter using primers that included 40 bp of homology immediately upstream of the start codon or 40 bp of homology downstream of the region used for homologous recombination for endogenous tagging of BCC0 (primers P15-P18).

For all tagging and knockout constructs, approximately 50 μg of the sequence-verified pU6-Universal plasmid and the PCR-amplified HDR template were electroporated into the appropriate parasite strain. Transfected cells were allowed to invade a confluent monolayer of HFFs overnight, and appropriate selection was applied. Successful tagging or knockout was confirmed by IFA, and clonal lines were obtained through limiting dilution. Knockout of BCC0 was verified by PCR using primers P19-P22.

## Generation of BCC0$^{cKD}$ strain

BCC0 was endogenously tagged with a 2xStrep3xTy epitope tag in an RHΔ*ku80*-Tati-HXGPRT parent strain as described above. To replace the endogenous promoter with an ATc-regulatable promoter, a pU6-Universal plasmid containing a protospacer against the 5' UTR of BCC0 approximately 250 bp upstream of the start codon was generated, as described previously [74,75]. The HDR template was PCR amplified from the pDHFR-TetO7-SAG4p vector using primers which included 40 bp of homology approximately 1 kb upstream of the start codon and 40 bp of homology immediately downstream of the start codon. Primers that were used for the pU6-Universal plasmid and the HDR template are listed in S3 Table (primers P23-P26). Clones were screened by IFA -/+ ATc (1 μg/mL). A clonal strain that was Ty-positive in the -ATc condition and Ty-negative in the +ATc condition was isolated and designated as BCC0$^{cKD}$.

## Knock-in of BCC0 complementation constructs

The BCC0 endogenous promoter (EP) was amplified from genomic DNA using primers P27 and P28. The BCC0 coding region was PCR amplified from cDNA using primers P29 and P30. The smHA tag was PCR amplified from psmHA.LIC-DHFR using primers P31 and P32. The entire plasmid except for the IMC32 promoter, coding region, and 3xHA tag was amplified from pUPRTKO-EP-IMC32$^{WT}$-3xHA using primers P33 and P34 [39]. The four fragments were ligated using Gibson assembly to create pUPRTKO-EP-BCC0$^{WTc}$-smHA (BCC0$^{WTc}$). This complement vector was then linearized with PsiI-HFv2 and transfected into BCC0$^{smOL-LAS}$ or BCC0$^{cKD}$ parasites along with a pU6 that targets the UPRT coding region. Selection was performed with 5 μg/mL 5-fluorodeoxyuridine (FUDR) for replacement of UPRT. Clones were screened by IFA, and an smHA-positive clone was isolated. BCC0$^{WTc}$ was used as the template to generate all deletion constructs using Q5 site-directed mutagenesis with primers P35-P64 (E0552S, NEB). The same processes for linearization, transfection, and selection were

followed for all deletion and mutant constructs. All restriction enzymes were purchased from NEB.

### Immunofluorescence assay

Confluent HFF cells were grown on glass coverslips and infected with *T. gondii*. After 24 hours, the coverslips were fixed with 3.7% formaldehyde in PBS and processed for immunofluorescence as described [77]. Primary antibodies were detected by species-specific secondary antibodies conjugated to Alexa Fluor 594/488/405 (ThermoFisher). Coverslips were mounted in Vectashield (Vector Labs), viewed with an Axio Imager.Z1 fluorescent microscope, and processed with ZEN 2.3 software (Zeiss).

### Western blot

Parasites were lysed in 1x Laemmli sample buffer with 100 mM DTT and boiled at 100°C for 5 minutes. Lysates were resolved by SDS-PAGE and transferred to nitrocellulose membranes, and proteins were detected with the appropriate primary antibody and corresponding secondary antibody conjugated to horseradish peroxidase. Chemiluminescence was induced using the SuperSignal West Pico substrate (Pierce) and imaged on a ChemiDoc XRS+ (Bio-Rad). Signal intensity was quantified using ImageLab. The adjusted volume of the Ty band (detecting BCC0$^{cKD}$) relative to the adjusted volume of the corresponding catalase loading control band was plotted. Raw data for western blot quantification is shown in S4 Table.

### Plaque assay

HFF monolayers were infected with 100–400 parasites and allowed to form plaques for 7 days. Cells were then fixed with ice-cold methanol and stained with crystal violet. All plaque assays were performed in triplicate. To quantify plaque number, the total number of plaques in each condition was counted manually. Plaque efficiency was calculated by dividing the number of plaques formed by the number of parasites infected in each replicate. The number of plaques or plaque efficiency for each replicate was plotted, and error bars were used to show standard deviation. To quantify plaque size, the areas of 30–50 plaques per condition were measured using ZEN software (Zeiss). BCC0$^{cKD}$ and BCC0$^{cKD}$ + BCC0$^{\Delta701-877}$ parasites that were treated with ATc exhibited a severe defect in plaque efficiency (<8%). For these experiments, all plaques formed were measured. The mean plaque size for each replicate was plotted, and error bars were used to show standard deviation. Graphical and statistical analyses were performed using Prism GraphPad 8.0. Raw data for plaque assays is shown in S4 Table.

### Affinity capture of biotinylated proteins

For affinity capture of proteins from whole cell lysates, HFF monolayers infected with IMC32$^{TurboID}$ or control parasites (RHΔ*hxgprt*Δ*ku80*, WT) were grown in normal media for 25 hours. Then, the media was supplemented with 150 μM biotin and parasites were allowed to grow for an additional 5 hours. Intracellular parasites in large vacuoles were collected by manual scraping, washed in PBS, and lysed in radioimmunoprecipitation assay (RIPA) buffer (50 mM Tris [pH 7.5], 150 mM NaCl, 0.1% SDS, 0.5% sodium deoxycholate, 1% NP-40) supplemented with Complete Protease Inhibitor Cocktail (Roche) for 30 min on ice. Lysates were centrifuged for 15 min at 14,000 x g to pellet insoluble material, and the supernatant was incubated with Streptavidin Plus UltraLink resin (Pierce) overnight at 4°C under gentle agitation. Beads were collected and washed five times in RIPA buffer, followed by three washes in 8 M urea buffer (50 mM Tris-HCl [pH 7.4], 150 mM NaCl) [78]. Samples were submitted for on-

bead digests and subsequently analyzed by mass spectrometry. The experiment was performed in duplicate.

## Mass spectrometry of biotinylated proteins

Purified proteins bound to streptavidin beads were reduced, alkylated, and digested by sequential addition of Lys-C and trypsin proteases. Samples were then desalted using C18 tips (Pierce) and fractionated online using a 75-μm inner-diameter fritted fused silica capillary column with a 5-μm pulled electrospray tip and packed in-house with 25 cm of C18 (Dr. Maisch GmbH) 1.9-μm reversed-phase particles. The gradient was delivered by a 140-minute gradient of increasing acetonitrile and eluted directly into a Thermo Orbitrap Fusion Lumos instrument where MS/MS spectra were acquired by Data Dependent Acquisition (DDA). Data analysis was performed using ProLuCID and DTASelect2 implemented in Integrated Proteomics Pipeline IP2 (Integrated Proteomics Applications) [79–81]. Database searching was performed using a FASTA protein database containing *T. gondii* GT1-translated open reading frames downloaded from ToxoDB. Protein and peptide identifications were filtered using DTASelect and required a minimum of two unique peptides per protein and a peptide-level false positive rate of less than 5% as estimated by a decoy database strategy. Candidates were ranked by spectral count comparing IMC32[TurboID] versus control samples [82]. The results were filtered to include only genes that had at least a two-fold enrichment and a difference of at least 5 when comparing the average spectral counts identified in the IMC32[TurboID] and control samples.

## Yeast two-hybrid

Y2H screening was performed by Hybrigenics Services as previously described [44,45]. Briefly, the full-length coding sequence of IMC32[C7S] was cloned into the pB27 vector (N-LexA-bait-C fusion) and transformed in yeast. This construct was screened for interactions against the *T. gondii* RH strain cDNA library with 23 million interactions tested. Confidence for each interaction was assessed algorithmically (Predicted Biological Score, PBS). Results were filtered to exclude any interactions with antisense or out-of-frame prey proteins.

For pairwise Y2H assays, fragments of BCC0, IMC32, and IMC43 were cloned into the pB27 (N-LexA-bait-C fusion) or pP6 (N-GAL4[AD]-prey-C fusion) vectors (Hybrigenics Services) as N-terminal fusions with the LexA DNA binding domain or GAL4 activation domain, respectively. All pB27 and pP6 constructs were cloned by Gibson Assembly or Q5 site-directed mutagenesis using primers P65-P134. Pairs of pB27 and pP6 constructs were co-transformed into the L40 strain of *S. cerevisiae* [MATa his3D200trp1-901 leu2-3112 ade2 LYS2::(4lexAop-HIS3) URA3::(8lexAop-lacZ) GAL4]. Strains were grown overnight in permissive (-Leu/-Trp) medium, normalized to $OD_{600} = 2$, then spotted in six serial dilutions onto permissive (-Leu/-Trp) and restrictive (-Leu/-Trp/-His) media. Growth was assessed after 3–5 days. Auto-activation was tested by co-transforming each pB27 fusion protein with an empty pP6 vector and co-transforming each pP6 fusion protein with an empty pB27 vector and performing spot assays for each strain as described above.

## Bioinformatic analysis of protein features

Coiled-coil domains were predicted using DeepCoil2 using a probability cut-off of 0.5 [50]. Palmitoylation sites were predicted using CSS-Palm 4.0 using a score cut-off of 5 [49]. Myristoylation sites were predicted using GPS-Lipid using a score cut-off of 5 [48]. Alpha helices and beta strands were predicted using PSIPRED using a cut-off of 5 and a minimum length of 4 amino acids for alpha helices or 3 amino acids for beta strands [57].

## Supporting information

**S1 Fig. The AID system does not provide a sufficient knockdown for analysis of BCC0 function.** A) Diagram and IFA of BCC0$^{AID}$ parasites grown for 24 hours -/+ IAA showing that BCC0$^{AID}$ does not provide sufficient protein knockdown (arrow). Magenta = anti-HA detecting BCC0$^{AID}$, Green = anti-IMC6. B) Diagram and IFA of BCC0$^{IAA7}$ parasites grown for 24 hours -/+ IAA showing that BCC0$^{IAA7}$ does not provide sufficient protein knockdown (arrow). Magenta = anti-HA detecting BCC0$^{IAA7}$, Green = anti-IMC6. Scale bars = 2 μm.
(TIF)

**S2 Fig. Conservation and secondary structure predictions for BCC0.** The amino acid sequence of BCC0 (TGGT1_294860) was aligned to its *N. caninum* ortholog NCLIV_001740 using ClustalO 1.2.4. Predicted features are shown above their corresponding sequences. Regions chosen for the deletion series are highlighted.
(PDF)

**S3 Fig. Most BCC0 deletion constructs localize normally.** IFAs showing that all BCC0 deletions except for BCC0$^{Δ701−877}$ (shown in Fig 4D) colocalize with endogenous BCC0$^{smOLLAS}$. Magenta = anti-HA detecting smHA-tagged BCC0 deletion constructs, Green = anti-OLLAS detecting endogenous BCC0$^{smOLLAS}$. Scale bars = 2 μm.
(TIF)

**S4 Fig. Validation of BCC0 knockout for strains depicted in Fig 4.** A) PCR verification for genomic DNA of BCC0$^{smOLLAS}$ (wild-type parent strain) and complemented Δ*bcc0* parasites. Diagram shows the binding location of primers used to amplify the BCC0 coding sequencing (blue arrows) and the site of recombination for the knockout (red arrows). The strain used in each PCR verification is indicated on the left of each image. B) IFA of complemented Δ*bcc0* parasites confirms loss of BCC0$^{smOLLAS}$ signal. Each IFA in panel B corresponds with the PCR verification to the left of it in panel A. Magenta = anti-OLLAS, Green = anti-IMC6. Scale bars = 2 μm.
(TIF)

**S5 Fig. ATc treatment is toxic to wild-type *T. gondii*.** Quantification of plaque size for RHΔ*ku80* parasites grown for seven days -/+ ATc. Statistical significance was determined using a two-tailed t-test (*, P < 0.05).
(TIF)

**S6 Fig. Localization and PCR verification for strains depicted in Fig 6.** A) IFAs showing that BCC0$^{170-899}$, BCC0$^{376-899}$, BCC0$^{570-899}$, and BCC0$^{701-899}$ all mislocalize. Magenta = anti-HA detecting smHA-tagged BCC0 deletion constructs, Green = anti-OLLAS detecting endogenous BCC0$^{smOLLAS}$. Scale bars = 2 μm. E) PCR verification for genomic DNA of complemented Δ*bcc0* parasites. Diagram indicates the binding location of primers used to amplify the BCC0 coding sequencing (blue arrows) and the site of recombination for the knockout (red arrows). A control PCR verification performed on BCC0$^{smOLLAS}$ (wild-type) parasites can be seen in S4A Fig.
(TIF)

**S1 Table. Full IMC32$^{TurboID}$ mass spectrometry results.** Full list of genes identified by mass spectrometry in the IMC32$^{TurboID}$ experiment. Spectral counts are shown for each gene. "Enrichment Diff" refers to the difference between the average spectral count in IMC32$^{TurboID}$ and control parasites. "Enrichment Fold" refers to the average spectral count for IMC32$^{TurboID}$ samples divided by the average spectral count for control samples. The second sheet labeled

"Filtered Results" shows the 1,117 genes that were at least two-fold enriched with a difference of at least five spectral counts.
(XLSX)

**S2 Table. Full IMC32 Y2H screen results.** Full list of binding interactions identified by the Hybrigenics IMC32 Y2H screen. All clones identified for a specific gene are grouped. Clones that were found to be out-of-frame or antisense are greyed out. Global PBS indicates the confidence score assigned to each clone [44,45]. The second sheet labeled "Hits" shows the 15 genes that were identified.
(XLSX)

**S3 Table. Oligonucleotides used in this study.**
(XLSX)

**S4 Table. Raw data for quantification of western blot and plaque assays.**
(XLSX)

## Acknowledgments

We thank Dominique Soldati-Favre for the catalase antibody and Michael Reese for the pP6 and pB27 plasmids, L40 strain of yeast, and protocols for pairwise Y2H assays.

## Author Contributions

**Conceptualization:** Rebecca R. Pasquarelli, Peter J. Bradley.

**Data curation:** Jihui Sha, James A. Wohlschlegel.

**Formal analysis:** Rebecca R. Pasquarelli.

**Funding acquisition:** James A. Wohlschlegel, Peter J. Bradley.

**Investigation:** Rebecca R. Pasquarelli, Jihui Sha.

**Project administration:** James A. Wohlschlegel, Peter J. Bradley.

**Resources:** James A. Wohlschlegel, Peter J. Bradley.

**Supervision:** James A. Wohlschlegel, Peter J. Bradley.

**Validation:** Rebecca R. Pasquarelli.

**Visualization:** Rebecca R. Pasquarelli.

**Writing – original draft:** Rebecca R. Pasquarelli, Jihui Sha, Peter J. Bradley.

**Writing – review & editing:** Rebecca R. Pasquarelli, Peter J. Bradley.

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
