## [Decision Letter · Decision Letter 0]

17 Jun 2024

Dear Peter,

Thank you very much for submitting your manuscript "BCC0 collaborates with IMC32 and IMC43 to form the Toxoplasma gondii essential daughter bud assembly complex" for consideration at PLOS Pathogens. As with all papers reviewed by the journal, your manuscript was reviewed by members of the editorial board and by several independent reviewers. The reviewers appreciated the attention to an important topic. Based on the reviews, we are likely to accept this manuscript for publication, providing that you modify the manuscript according to the review recommendations.

You will see that one of the experts who reviewed your manuscript mentions the somewhat incremental nature of the work, which is something I initially quite agreed with. On the other hand, all three reviewers found that it is a thorough and very complete study, and thus in the end I feel that, overall, the insightful information it brings about the assembly of the inner membrane complex would justify publication in PLOS Pathogens. However, before the manuscript can be formally accepted you should address carefully all the points raised by the reviewers. I also ask you to please consider the following additional points:

- at the end of the Summary (l. 54-55), as the present work does not provide direct insights into the development of new anti-toxoplasma drugs, I think that the “which could lead to the identification of drug targets” part should be removed.

- on l. 193 you mention a "93% reduction in the level of BCC0 expression", which implies that some quantifications have been performed, yet they are not shown in the manuscript. Either show them or remove this particular number.

Sincerely,

Sébastien Besteiro, PhD

Academic Editor

PLOS Pathogens

Meera Nair

Section Editor

PLOS Pathogens

Michael Malim

Editor-in-Chief

PLOS Pathogens

orcid.org/0000-0002-7699-2064

Reviewer Comments (if any, and for reference):

Reviewer's Responses to Questions

**Part I - Summary**

Reviewer #1: This study reports on dissecting the organization and function of a complex of three early recruited daughter proteins which are essential for the proper assembly of the IMC during T. gondii tachyzoite endodyogeny. The work is thorough and complete in all aspects, making this a very robust report on how this complex interacts with each other, and which sections of BCC0 are essential to its function. The elegant domain dissection using a series of deletion constructs in different background strains highlights only the domain identified by Y2H for interaction with IMC32 is essential for localization and completion of the lytic cycle. As discussed by the authors, it is somewhat surprising none of the acylation sites exerted a critical function, but the data is clear and convincing and indeed as remarked, these are only predicted acylation sites. Lastly, the domains on both IMC32 and BCC0 that are interacting with each other were exhaustively mapped using pairwise Y2H with deletion mutants. As the authors discuss, many questions remain as to the regulation of this complex in space and time, but this study lines up the tools to tackle these questions next. In all, the study is a detailed report on a critical complex key to the initiation of daughter budding and therefore entails a major step forward in our understanding of this process unique to apicomplexan parasites

Reviewer #2: In this study, Pasquarelli et al. investigate the organization of the inner membrane complex (IMC) during the internal development of the daughter cells in Toxoplasma gondii. Capitalizing their previous work on IMC32 (PMID: 33593973) and IMC43 (PMID: 37782662), they dig further into the composition and function of the complex in which these two proteins are involved. Crossing data from a yeast two-hybrid screen and proximity labeling, they identified BCC0 as a third component of this “essential daughter bud assembly complex”. They convincingly show that BCC0 recruitment is dependent of IMC32 and IMC43. Using a complementation strategy with a large collection of BCC0 mutants, they pinpoint the region of the protein essential for both its localization and function. Surprisingly, only the construct lacking the region 701-877 has the same impact as the depletion of the full-length protein. A contribution of residues 170-375 was also observed and the authors then assessed how much of this fairly large protein is sufficient for its function. They show that the C-terminal part of the protein, downstream of residue 899, is dispensable.

Finally, using previous Y2H constructs of IMC32 and IMC43, the authors identified the regions involved in the direct interactions of the proteins and showed that BCC0 and IMC43 bind to IMC32 leading to the proposed model of IMC assembly with the hierarchical recruitment of IMC32, IMC43 and BCC0.

The manuscript is clear and well-written. The experiments are well-conducted and controlled. The IMC is a crucial component of the daughter cell assembly in Apicomplexa and its components not present in the host could constitute an attractive drug target. While these proteins are not conserved across the phylum but found in Coccidians, the manuscript is, to my view, of interest for the audience of PLOS pathogens. It contributes to the elucidation of the molecular assembly of daughter cell IMC. I just have a few minor concerns/corrections about the paper.

Reviewer #3: The manuscript by Pasquarelli and colleagues investigates the role of IMC32, IMC43, and BCC0 for Toxoplasma gondii inner membrane formation. Using proximity labeling and Y2H, the authors demonstrate a likely complex of IMC32, IMC43, and BCC0. Straightforward reverse genetic experiments demonstrate that BCC0 is essential and that 701-877 are critical for localization and function of the protein. The studies are well done, and the data are convincing. The manuscript provides a new level of knowledge about IMC biogenesis, an area of apicomplexan biology that is really lacking many details. Thus, the study provides an important, albeit somewhat incremental, step forward in our understanding of endodyogeny.

**Part II – Major Issues: Key Experiments Required for Acceptance**

Reviewer #1: 1. Fig 5E reports on plaque size, but another aspect was the sharp reduction in plaque number reported in Fig 3D. How did the complementation with delta701-877 affect the number of plaques? It would be great if this could be included.

Reviewer #2: (No Response)

Reviewer #3: Major points (although none are really major):

1. Fig 3D vs 5E plaque size for BCC0 cKD are somewhat different. Is this difference meaningful or merely showing the variability of biological assays? This should be stated either way.

2. It would be helpful to have a the cKD control included in figure 6J. This would be helpful so that a comparison can be made between the “partial” complementation and no complementation. Alternatively, it would be ideal to have some sort of inducible knockout for endogenous BCC0 with each of the complementation strains. That would allow for immediate removal of wt BCC0 without the ability for compensatory mutations to occur. This is mostly technical because the results are quite robust, as is.

3. It would be interesting for the authors to hypothesize or at least discuss how the BCC0 truncations, especially the one that is just 701-899, partially complements despite mislocalization. The discussion mentions the interaction with IMC32 as the driver. However, this seems like it is likely to allow the truncated protein to interact but not how it functions without 80+% of the protein. This is sort of present already in lines 300-318. However, the results still seem surprising.

**Part III – Minor Issues: Editorial and Data Presentation Modifications**

Reviewer #1: 1. Plaque assay quantifications. Although the M&M states 30-50 plaques were measured for size, it does not state whether biological reps were performed. In the Fig legends it does not provide details either. Please also add whether the mean/average/median is plotted and what metric the error bars represent. To be complete, inclusion of all quantitation and analysis data as a supplement is recommended.

2. Fig 1B. The apicoplast is known to harbor biotin, yet in the -biotin control panel no streptavidin signal is visible at all. Please comment.

Reviewer #2: I have only a few minor concerns about the paper:

1. As mentioned in the text (line 209), the authors used an alignment of the apicomplexan BCC0 and secondary structure predictions with PsiPred to establish all the constructs described in the study. It would be useful to see this sequence homology as a supplementary figure to better understand how the constructs have been chosen and how much the region between residues 1-899 is conserved.

2. The protein TGGT1_280370 is ranked A from the Y2H screen and is the 2nd protein enriched of the cross between the Y2H and the proximity labeling. Did the author assess its localization? It might be out of the scope of this suty.

3. Figure 4. The panel with the plaque area graph should be panel P and accordingly, line 227 “Figure 4M” should be “Figure 4P”.

4. In the proposed model, how to place IMC44 that interacts with IMC43?

Reviewer #3: Minor points:

1. Line 67 – this notes that current therapies for Toxoplasmosis are not well tolerated in humans. This is not really true because the first line medications, pyrimethamine with sulfadiazine, is actually very well tolerated when dosed properly. The statement does not add to the manuscript and is frankly incorrect.

2. Line 198-200. It would be helpful for the reader to know what the tags were for IMC32 and IMC43. This should be added to the text here.

PLOS authors have the option to publish the peer review history of their article (what does this mean?). If published, this will include your full peer review and any attached files.

Reviewer #1: No

Reviewer #2: No

Reviewer #3: No

Figure Files:

Data Requirements:

Reproducibility:

References:

---

## [Editor Report · Decision Letter 1]

3 Jul 2024

Dear Peter,

Thank you very much for submitting your manuscript "BCC0 collaborates with IMC32 and IMC43 to form the Toxoplasma gondii essential daughter bud assembly complex" for consideration at PLOS Pathogens.

Thank you for addressing the reviewers’ comments. I think that they have been satisfactorily addressed . However, one thing remains to be checked before the manuscript can be accepted: on new Fig. 5F, when comparing the -/+ conditions of the BCC0cKD+BCC0WTc cell line, given the datapoints and the size of the error bars, it is very unlikely that they would be significantly different with a p value < 0.05. Please verify and correct accordingly.

Sincerely,

Sébastien Besteiro, PhD

Academic Editor

PLOS Pathogens

Meera Nair

Section Editor

PLOS Pathogens

Michael Malim

Editor-in-Chief

PLOS Pathogens

orcid.org/0000-0002-7699-2064

One thing remains to be checked before the manuscript can be accepted: on new Fig. 5F, when comparing the -/+ conditions of the BCC0cKD+BCC0WTc cell line, given the datapoints and the size of the error bars, it is very unlikely that they would be significantly different with a p value < 0.05. Please verify and correct accordingly.

Reviewer Comments (if any, and for reference):

Figure Files:

Data Requirements:

Reproducibility:

References:

---

## [Editor Report · Decision Letter 2]

10 Jul 2024

Dear Prof Bradley,

We are pleased to inform you that your manuscript 'BCC0 collaborates with IMC32 and IMC43 to form the Toxoplasma gondii essential daughter bud assembly complex' has been provisionally accepted for publication in PLOS Pathogens.

Best regards,

Sébastien Besteiro, PhD

Academic Editor

PLOS Pathogens

Dominique Soldati-Favre

Section Editor

PLOS Pathogens

Michael Malim

Editor-in-Chief

PLOS Pathogens

orcid.org/0000-0002-7699-2064
---

## [Editor Report · Acceptance letter]

13 Jul 2024

Dear Prof Bradley,

We are delighted to inform you that your manuscript, "BCC0 collaborates with IMC32 and IMC43 to form the Toxoplasma gondii essential daughter bud assembly complex," has been formally accepted for publication in PLOS Pathogens.

Best regards,

Michael Malim

Editor-in-Chief

PLOS Pathogens

orcid.org/0000-0002-7699-2064